# Characteristic Length for Pinning Force Density in Nb_3_Sn

**DOI:** 10.3390/ma16145185

**Published:** 2023-07-24

**Authors:** Evgeny F. Talantsev, Evgeniya G. Valova-Zaharevskaya, Irina L. Deryagina, Elena N. Popova

**Affiliations:** 1M. N. Miheev Institute of Metal Physics, Ural Branch, Russian Academy of Sciences, 18, S. Kovalevskaya St., 620108 Ekaterinburg, Russia; valova@imp.uran.ru (E.G.V.-Z.); deryagina@imp.uran.ru (I.L.D.); popova@imp.uran.ru (E.N.P.); 2NANOTECH Centre, Ural Federal University, 19 Mira St., 620002 Ekaterinburg, Russia

**Keywords:** pinning force density in superconductors, superconducting critical current, scaling laws in superconductivity

## Abstract

The pinning force density, $F_{p}$, is one of the main parameters that characterize the resilience of a superconductor to carrying a dissipative-free transport current in an applied magnetic field. Kramer (1973) and Dew-Hughes (1974) proposed a widely used scaling law for this quantity, where one of the parameters is the pinning force density maximum, $F_{p,max}$, which represents the maximal performance of a given superconductor in an applied magnetic field at a given temperature. Since the late 1970s to the present, several research groups have reported experimental data on the dependence of $F_{p,max}$ on the average grain size, d, in Nb_3_Sn-based conductors. $F_{p,max}\left(d\right)$ datasets were analyzed and a scaling law for the dependence $\left|F_{p,max}\left(d\right)\right|=A\times ln\left(1/d\right)+B$ was proposed. Despite the fact that this scaling law is widely accepted, it has several problems; for instance, according to this law, at T=4.2 K and d≥650 nm, Nb_3_Sn should lose its superconductivity, which is in striking contrast to experiments. Here, we reanalyzed the full inventory of publicly available $F_{p,max}\left(d\right)$ data for Nb_3_Sn conductors and found that the dependence can be described by the exponential law, in which the characteristic length, δ, varies within a remarkably narrow range of δ=175±13 nm for samples fabricated using different technologies. The interpretation of this result is based on the idea that the in-field supercurrent flows within a thin surface layer (thickness of δ) near grain boundary surfaces (similar to London’s law, where the self-field supercurrent flows within a thin surface layer with a thickness of the London penetration depth, λ, and the surface is a superconductor–vacuum surface). An alternative interpretation is that δ represents the characteristic length of the exponential decay flux pinning potential from the dominant defects in Nb_3_Sn superconductors, which are grain boundaries.

## 1. Introduction

Multifilamentary Nb_3_Sn wires are utilized in many international mega-science projects, such as the Large Hadron Collider (LHC) [1] and International Thermonuclear Experimental Reactor (ITER) [2]. The advantages of using Nb_3_Sn-based superconductors are the high current-carrying capacity in high magnetic fields, low cost, and availability of at least three different technologies for device manufacturing. In particular, to create high-field large-aperture quadrupole MQXF [3] and high-field 11-T dipoles [4] for the high-luminosity LHC Upgrade Project, new generations of high-field Nb_3_Sn-based superconductors have been developed [5]. The critical current density $J_{c}$ of these modern Nb_3_Sn conductors (strands) achieved record values of non-Cu $J_{c}\left(B=12\;T,\;T=4.2\;K\right)=3000\;A/{\mathrm{mm}}^{2}$ and $J_{c}\left(B=15\;T,\;T=4.2\;K\right)=1700\;A/{\mathrm{mm}}^{2}$ [6]. According to [7], the creation of a Future Circular Collider (FCC) at CERN requires Nb_3_Sn-based wires with $J_{c}\left(B=16\;T,\;T=4.2\;K\right)=1500\;A/{\mathrm{mm}}^{2}$ or $J_{c}\left(B=12\;T,\;T=4.2\;K\right)=3500\;A/{\mathrm{mm}}^{2}$.

For the ITER project, bronze-processed Nb_3_Sn-based wires were developed for superconducting magnets, providing a critical current density $J_{c}$ of approximately 750 A/mm^2^ in an applied field of 12 T at liquid helium temperatures [2]. However, in [8], the same wire was processed and achieved a $J_{c}\left(12\;T,4.2K\right)$ of 1000 A/mm^2^, which demonstrates that further advancement of cable manufacturing technology from Nb_3_Sn strands to cables is desired.

The development of this new cable manufacturing technology is crucially important for the next mega-science project after the ITER, which is the DEMO experimental facility. The DEMO project requires superconducting Nb_3_Sn-based conductors with even higher current capacities [8].

Extensive (over nearly five decades) R&D studies of Nb_3_Sn-based conductors have shown that the key factors affecting the in-field critical current in these wires are the local composition, structure, and morphology of the superconducting A-15 phase [9,10,11,12,13,14,15,16,17].

These studies also showed that in high magnetic fields, the main pinning centers in Nb_3_Sn-based composites are grain boundaries, and the conventional approach to increasing$\;J_{c}\left(B,\;T\right)$ in Nb_3_Sn is to maximize the density of the grain boundaries, that is, to ensure grain refinement. To achieve this goal, various manufacturing methods and multifilamentary wire designs have been proposed [7] that target the creation of small average size of grains with low dispersion and high homogeneity [18,19,20,21].

Nb_3_Sn-based superconducting wires are produced by one of the following methods: bronze route, internal tin (IT), and power in tube (PIT) [22,23,24].

In our study, we analyzed the transport current characteristics of wires manufactured by the bronze route and the PIT method. Therefore, it is necessary to present a brief introduction to these methods.

In the bronze route [25], an initial billet is formed of Nb, Nb-Ti, or Nb-Ta rods assembled in a bronze Cu-Sn matrix and an external copper tube is extruded and drawn. Sn diffusion from the Cu-Sn matrix forms the Nb_3_Sn phase in Nb filaments under heat treatment (HT). HT is commonly known as diffusion annealing. The solid-state diffusion of Sn from the Cu-Sn matrix at relatively low temperatures prevents excessive grain growth and thus causes an increase in the magnetic flux pinning efficiency. One of the known disadvantages of this method is the limited solubility of Sn in Cu-Sn alloys. In addition, when the Sn concentration is increased to more than 8 mass.%, the alloy becomes brittle, due to the precipitated ε (Cu_3_Sn) phase. This alloy prevents plastic deformation and leads to the cracking of the composite wire during processing. Therefore, to ensure that a sufficient amount of Sn can be yielded to form the Nb_3_Sn phase, the minimum desirable ratio of the volume fractions of bronze and niobium should be approximately 3:1. As a result, the effective portion of Nb_3_Sn in the entire conductor is low, and thus, bronze-processed wires have lower *J*_e_ values in comparison with other methods (IT, PIT) that provide a higher Nb_3_Sn volume phase ratio. In addition, this technology requires frequent in-process annealing during wire drawing to avoid cracking the bronze matrix.

However, Abacherli et al. [26,27] advanced the bronze route technology by introducing the Swissmetal (Dornach, Switzerland) Osprey-processed bronze with 15.4 wt.% tin content and Nb7.5wt.%Ta as core materials for multifilamentary (Nb,Ta)_3_Sn wires. This technology was later introduced for tantalum-free Nb_3_Sn-based multifilamentary wires. This technology is known as the Osprey process within bronze manufacturing technologies. Using this technology, it is possible to increase the number of Nb filaments in the strand and provide a complete transformation of the Nb filaments into the superconducting phase. In addition, this increases the Sn concentration in the Nb_3_Sn layers, resulting in increases in *J*_c_ and *J*_e_ [26].

However, even in Nb_3_Sn strands fabricated using the Osprey-processed bronze matrix, it is not possible to avoid large Nb_3_Sn composition gradients across the superconducting layer. These gradients produce large gradients in the superconducting properties that limit the overall in-field transport current density [9]. As shown in reference [27], this tin deficiency causes the formation of a relatively large fraction of non-stoichiometric Nb_3_Sn compounds. It should be noted that the Nb_3_Sn phase is stable at 18–25 at.% Sn, and the superconducting parameters, including in-field current density, of the Nb_3_Sn are degraded versus decreasing tin content [28].

The second widely used technology for manufacturing multifilamentary Nb_3_Sn wires is the IT process [29]. This technology was developed to avoid frequent in-process annealing, which is an essential component of the bronze route. This method utilizes separate Sn, Cu, and Nb billet stacking elements, which enhance the Sn concentration in the matrix in comparison to the bronze process [30]. As a result, modern IT strands (e.g., strands with distributed diffusion barriers) exhibit *J_c_* values above 2200 A/mm^2^ and achieve a record-breaking value of 3000 A/mm^2^ (non-Cu, l2 T, 4.2 K) [12,31]. It should be noted that in the literature, non-Cu *J_c_* refers to the transport critical current over the cross-section of the conductor without the stabilizing copper layer. Mentioned above, the highest critical current densities refer to the non-Cu *J_c_*. The highest non-Cu *J_c_* values achieved for multifilamentary strands (made by the IT process) originate from the high chemical and microstructural homogeneity and the high fraction of the stoichiometric Nb_3_Sn phase.

A new approach to increasing the *J_c_* of superconductors, called the Restacked Rod Process (RRP) [32], is based on IT technology. Because the dependence of the pinning force density versus Nb_3_Sn grain size for wires fabricated by this technology is still unavailable in the public domain, we do not discuss this process herein, and refer the readers for details of this process to references [33,34].

The third technology for the fabrication of multifilamentary Nb_3_Sn wires with a relatively high current density (>2500 A/mm^2^) is the PIT process [35]. This method combines a Sn-rich source and fine filaments (approximately 35 μm), resulting in PIT wires containing a relatively large volume fraction of the A15 phase, which is close to the stoichiometric intermetallic compound.

There are many advantages of the PIT process, such as shorter heat treatments (owing to the close location of the Sn source to the niobium), no pre-heating treatment, and relatively small filaments (30–50 µm) that can be used for manufacturing. The latter leads to low hysteresis losses in the conductor. However, the main disadvantage of the PIT manufacturing routine is the high cost compared the two other main fabrication technologies for Nb_3_Sn wires [36,37].

The resilience of any superconducting wire to carrying a dissipative-free transport current at an applied magnetic field can be quantified by the pinning force density, $\vec{F_{p}}$ (defined as the vector product of the transport critical current density,$\;\vec{J_{c}}$, and the applied magnetic field, $\vec{B}$):(1)$$\vec{F_{p}}\left(J_{c},B\right)=\vec{J_{c}}⊗\vec{B}.$$

For an isotropic superconductor and maximal Lorentz force geometry, that is, when $\vec{J_{c}}⊥\vec{B}$, Kramer [38] and Dew-Hughes [39] proposed a widely used scaling expression for the amplitude of the pining force density [40]:(2)$$\left|\vec{F_{p}}\left(B\right)\right|=F_{p,max}\times \frac{{\left(p+q\right)}^{p+q}}{p^{p}q^{q}}\times {\left(\frac{B}{B_{c2}}\right)}^{p}\times {\left(1-\frac{B}{B_{c2}}\right)}^{q},$$
where $F_{p,max}$, $B_{c2}$, *p*, and *q* are free-fitting parameters, $B_{c2}$ is the upper critical field, and $F_{p,max}$ is the pinning force density amplitude.

Figure 1 shows a typical $\left|\vec{F_{p}}\left(B,4.2\;K\right)\right|$ for Nb_3_Sn superconductors reported by Flükiger et al. [41], where the data fit to Equation (2) and the deduced free-fitting parameters, $F_{p,max}$, $B_{c2}$, *p*, and *q*, are shown.

While the upper critical field, $B_{c2}$, is one of the fundamental parameters for a given superconducting phase, three other parameters in Equation (2), namely $F_{p,max}$, *p*, and *q*, depend on the superconductor microstructure, the presence of secondary phases, etc. In accordance with the approach proposed by Dew-Hughes [39], the shape of $\left|\vec{F_{p}}\left(B\right)\right|$ (defined by *p* and *q*) reflects the primary pinning mechanism in a sample. Dew-Hughes [39] calculated the theoretical characteristic values of *p* and *q* for different pinning mechanisms, particularly for point defect (PD) and grain boundary (GB) pinning.

The evolution of the dominant pinning mechanism from GB- to PD-pinning in Nb_3_Sn under neutron irradiation was recently reported by Wheatley et al. [42], who showed that the unirradiated Nb_3_Sn alloy exhibits the $\left|\vec{F_{p}}\left(B,T\right)\right|$ form, indicating the dominance of GB-pinning, and after the neutron irradiation, the $\left|\vec{F_{p}}\left(B,T\right)\right|$ form transforms towards the PD-pinning mode.

It should be noted that to extract the partial contribution of GB- and PD-pinning from the total pinning of the Nb_3_Sn wire, Tarantini et al. [43] presented the total $\left|\vec{F_{p}}\left(B\right)\right|$ as a sum of two terms with fixed *p* and *q* values for GB- and PD-pinning, where introduced $a_{GB}$ and $a_{PD}$ designated as amplitudes for GB- and PD-pinning, respectively.

The fourth parameter in Equation (2), which is the $F_{p,max}$, represents the maximal performance of a given superconductor in an applied magnetic field. It is a well-established experimental fact [41,44,45,46,47,48,49,50] that the $F_{p,max}$ in Nb_3_Sn depends on the average grain size, d, of the material. The traditional approach to representing the $F_{p,max}$ vs. d dependence is to use a reciprocal semi-logarithmic plot (Figure 2). Godeke [45] proposed the following form for the $F_{p,max}$ vs. d dependence:(3)$$F_{p,max}\left(d\right)=A\times ln\left(1/d\right)+B,$$
where free-fitting parameters A=22.7 and B=−10.

Following traditional methodology [40], Godeke [45] proposed that because grain boundaries are the primary pinning centers in Nb_3_Sn, there is an optimum grain size, $d_{opt}$, at which the maximum performance for a given wire can be achieved for a given applied magnetic field, *B*. This field [45] is equal to the flux line spacing in the hexagonal vortex lattice, $a_{hexagonal}$ [51], at the applied field B, which can be designated as the matching field, $B_{match}$, at the maximum pinning force density:(4)$$d_{opt}=a_{hexagonal}={\left(\frac{4}{3}\right)}^{1/4}\times {\left(\frac{\phi_{0}}{B_{match}}\right)}^{1/2},$$
where $\phi_{0}=\frac{h}{2e}$ is the superconducting flux quantum.

Here, we show that neither Equation (3) nor Equation (4) provides a valuable description of the available experimental $F_{p,max}\left(d\right)$ data measured over several decades in Nb_3_Sn conductors. We also propose a new model to describe a full set of publicly available experimental datasets on the maximum pinning force density vs. grain size, $F_{p,max}\left(d\right)$.

## 2. Problems Associated with Current Models

Equation (4) implies that if the grain size, $d_{opt}$, in some Nb_3_Sn conductors has been determined, then the matching applied magnetic field, $B_{match}$, can be calculated from Eq. 4. Following this logic [45], one can expect that the maximal performance in magnetic flux pinning, namely $F_{p,max}$, should be observed at $B_{match}$:(5)$$B_{match}\left(d_{opt}\right)=B_{F_{p,max}}\left(d_{opt}\right)={\left(\frac{4}{3}\right)}^{1/2}\times \left(\frac{\phi_{0}}{d_{opt}{}^{2}}\right).$$

In Figure 1, we fitted the $\left|\vec{F_{p}}\left(B\right)\right|$ data [41] to Equation (1) for Nb_3_Sn conductors with different grain sizes, d, from which $B_{F_{p.max},exp}\left(d\right)$ values were extracted. In Figure 3, we show $B_{F_{p.max},exp}\left(d\right)$ and calculated $B_{F_{p.max},calc}\left(d\right)$ (Equation (5)), from which it can be concluded that the traditional understanding of the primary mechanism governing dissipative-free high-field current capacity in Nb_3_Sn conductors [45] is incorrect, and there is a quest to understand the main mechanisms that determine the maximal in-field performance of Nb_3_Sn wires. However, the solution to the problem cannot be based on the idea that there is some optimal spatial separation of vortices (or, in other words, optimal magnetic flux density) for a given average grain size [45], because this assumption contradicts the data shown in Figure 3. Thus, there is a need to determine the primary mechanisms for obtaining the maximal in-field performance of Nb_3_Sn wires.

The validity of the $F_{p,max}\left(d\right)$ scaling law proposed by Godeke (Equation (3) [45]) was analyzed and it was concluded that there are at least three fundamental problems with the law:The logarithmic function used in Equation (3), as well as all other mathematical functions, can operate only with dimensionless variables, whereas the variable in Equation (3) has the dimension of inverse length. For instance, the variable B in the Kramer–Dew-Hughes scaling law (Equation (2)) has the dimension cancelation term $\frac{1}{B_{c2}}$. The same general approach can be found for all equations in Ginzburg–Landau [51], Bardeen–Cooper–Schrieffer [52], and other physical theories [53], all of which implement this general rule.

For instance, the lower critical field, $B_{c1}$, in superconductors has a traditional form [54]:(6)$$B_{c1}\left(T\right)=\frac{\phi_{0}}{4\pi \lambda^{2}\left(T\right)}\times \left(\ln\left(\frac{\lambda \left(T\right)}{\xi \left(T\right)}\right)+\alpha \left(\frac{\lambda \left(T\right)}{\xi \left(T\right)}\right)\right),$$
where
(7)$$\alpha \left(\kappa \right)=\alpha_{\infty }+e^{\left(-c_{0}-c_{1}\times ln\left(\frac{\lambda \left(T\right)}{\xi \left(T\right)}\right)-c_{2}\times {\left(ln\left(\frac{\lambda \left(T\right)}{\xi \left(T\right)}\right)\right)}^{2}\right)}\pm \varepsilon ,$$
where $\lambda \left(T\right)$ is the London penetration depth, $\xi \left(T\right)$ is the superconducting coherence length, $\alpha_{\infty }=0.49693$, $c_{0}=0.41477$, $c_{1}=0.775$, $c_{2}=0.1303$, and ε≤0.00076. Equations (6) and (7) were recently simplified to the following form [55]:(8)$$B_{c1}\left(T\right)=\frac{\phi_{0}}{4\pi \lambda^{2}\left(T\right)}\times \left(\ln\left(1+\sqrt{2}\frac{\lambda \left(T\right)}{\xi \left(T\right)}\right)\right),$$

In Equations (6) and (8), the variable under the logarithm is dimensionless. The same can be found in the equation for the universal self-field critical current density, $J_{c}\left(sf,T\right)$, in thin film superconductors [56]:(9)$$J_{c}\left(T\right)=\frac{\phi_{0}}{4\pi \mu_{0}\lambda^{3}\left(T\right)}\times \left(\ln\left(\frac{\lambda \left(T\right)}{\xi \left(T\right)}\right)+0.5\right),$$
where $\mu_{0}$ is the permeability of the free space. It should be noted that Equation (9) was recently confirmed by Paturi and Huhtinen [57] for YBa_2_Cu_3_O_7−σ_ thin films that exhibit different mean-free paths for charge carriers.

The same principle was implemented in all general physics laws. For instance, diffusion laws are the primary laws that determine the formation of the Nb_3_Sn phase in multifilamentary wires [24]. In particular, we consider the diffusion coefficient, $D\left(T\right)$ [24]:(10)$$D\left(T\right)=D_{0}\times e^{-\frac{Q}{RT}},$$
where $D\left(T\right)$ is the diffusion coefficient, $D_{0}$ is the maximal diffusion coefficient, $D\left(T\right)$ and $D_{0}$ have the same units of $m^{2}\times s^{-1}$; the activation energy, Q, has units of $J\times {\mathrm{mol}}^{-1}$; the universal gas constant, Q, has unit of $J\times {\mathrm{mol}}^{-1}\times K^{-1}$; and absolute temperature, T, has units of K. Consequently, the variable under the exponential function is unitless.

Based on the above, Equation (3) should be transformed into a form that does not have a fundamental problem based on the use of the $ln\left(1/d\right)$ term. Following the form of other physical laws (see, for instance, Equations (7)–(10)), Equation (3) can be rewritten as:(11)$$F_{p,max}\left(d\right)=A\times ln\left(1/d\right)-B=ln\left[\frac{e^{-B}}{d^{A}}\right]=ln{\left[\frac{e^{-\frac{B}{A}}}{d}\right]}^{A}=A\times ln\left[\frac{D}{d}\right],$$
where $D=e^{-\frac{B}{A}}$, and after the substitution of A=22.7 and B=−10, one can obtain D=0.65, which following the logic above should have units of μm.

Nevertheless, Equation (11) formally has the correct mathematical form. However, it does not change the curve itself in Figure 2 and Figure 4, and, thus, two problems with Equations (3) and (11), which are in striking disagreement with the experiments, remain.
2.The first problem is the limit of Equations (3) and (11) for large grain sizes. In Figure 4, we replotted $\left|F_{p,max}\left(d\right)\right|$ data from Figure 2 in a linear–linear plot and showed both side extrapolations of Equations (3) and (11) within the range of 20 nm≤d≤800 nm, which is the usual range of grain sizes in Nb_3_Sn conductors. In Figure 2 and Figure 4, one can see that:

(12)$${\left|F_{p,max}\left(d\right)\right|}_{\;d\geq D=650\;\mathrm{nm}}={\left.\left(A\times ln\left(1/d\right)+B\right)\right|}_{d\geq D=650\;\mathrm{nm}}\leq 0,$$
which is a prohibited inequality in mathematics.

From a physical point of view, Equation (12) indicates that at d=D=650 nm, Nb_3_Sn loses its superconducting properties, that is, it converts to a normal state. Truly, by definition, $\left|F_{p,max}\left(d\right)\right|$ is the global maximum of the pinning force density for a given superconductor at a given temperature and any applied field (it should be noted that $\left|F_{p,max}\right|$ is achieved at $B=B_{Fp,max}$). If this value is equal to zero, then $\left|F_{p}\left(d\right)\right|$ for this superconductor at any other field $B\neq B_{Fp,max}$ is also equal to zero. This implies that there is no superconducting state at T=4.2 K for any applied field for Nb_3_Sn with grain sizes d≥D=650 nm, which is in striking disagreement with the experiment.

We also need to note that the free-fitting parameters deduced by us (A=21.9±1.2, B=−9.9±2.7) from the fit of the $\left|F_{p,max}\left(d\right)\right|$ dataset to Equations (3) and (11) are different from the values reported by Godeke [45], A=22.7, B=−10, who analyzed the same $\left|F_{p,max}\left(d\right)\right|$ dataset.
3.Another validity problem with Equations (3) and (11) is for small grain sizes:

(13)$$\underset{d\to 0}{\lim}\left|F_{p,max}\left(d\right)\right|=\underset{d\to 0}{\lim}\left(A\times ln\left(1/d\right)+B\right)=\infty ,$$ which is unphysical, because when d becomes comparable to the double coherence length (which is the size of a normal vortex core):(14)$$d_{\min}\left(4.2\;K\right)≅2\times \xi \left(T\right)=2\times \frac{\xi \left(0\right)}{\sqrt{1-\frac{T}{T_{c}}}}=\frac{2\times 3.0\;\mathrm{nm}}{\sqrt{1-\frac{4.2\;K}{18\;K}}}=6.9\;\mathrm{nm},$$
where $\xi \left(0\right)=3.0\;\mathrm{nm}$ [58] and $T_{c}=18\;K$ [58], a further decrease in the grain size d should not cause any changes in the magnetic flux pinning, and thus in the $\left|F_{p,max}\left(d\right)\right|$ amplitude.

## 3. Results

By experimenting with many analytical functions that can approximate the $\left|F_{p,max}\left(d\right)\right|$ dependence shown in Figure 2 and Figure 4, we found a remarkably simple, robust, heuristic, and physically sound expression:(15)$$\left|F_{p,max}\left(d\right)\right|=\left|F_{p,max}\left(0\right)\right|\times e^{-\frac{d}{\delta }},$$
where $\left|F_{p,max}\left(0\right)\right|$ and δ are free-fitting parameters. This function exhibits physically sound limits:(16)$$\underset{d\to \infty }{\lim}\left|F_{p,max}\left(d\right)\right|=\underset{d\to \infty }{\lim}\left(\left|F_{p,max}\left(0\right)\right|\times e^{-\frac{d}{\delta }}\right)=0,$$
(17)$$\underset{d\to 0}{\lim}\left|F_{p,max}\left(d\right)\right|=\underset{d\to 0}{\lim}\left(\left|F_{p,max}\left(0\right)\right|\times e^{-\frac{d}{\delta }}\right)=\left|F_{p,max}\left(0\right)\right|<\infty .$$

We propose interpretations for $\left|F_{p,max}\left(0\right)\right|$ and of δ parameters in Section 4. Before that, in this section, we show the robustness of Equation (15) for fitting publicly available datasets for Nb_3_Sn conductors. Data fitting was performed in OriginPro 2017 software.

### 3.1. Bronze Technology Samples

Bronze technology for Nb_3_Sn-based wires has been described in detail elsewhere [1]. For our analysis, we used the $\left|F_{p,max}\left(d\right)\right|$ dataset reported by Godeke [45]. Godeke [59] pointed out that Fischer [44] collected raw $\left|F_{p,max}\left(d\right)\right|$ data (shown in Figure 2 and Figure 4), and that these data are “*all pre-2002 results*” and that this dataset includes Fischer’s [45] “*non-Cu area*” data.

In Figure 5, we fitted this largest publicly available dataset for Nb_3_Sn conductors fabricated using bronze technology to Equation (15). The deduced parameters were $\left|F_{p,max}\left(0\right)\right|=74\pm 3\;\frac{\mathrm{GN}}{m^{3}}$ and δ=175±12 nm. The parameters have low dependence (~0.87), which indicates that our model (Equation (15)) is not over-parameterized.

### 3.2. Powder-in-Tube Technology Samples

Powder-in-tube technology for Nb_3_Sn-based wires has been described in detail elsewhere [1]. For our analysis, we used the $\left|F_{p,max}\left(d\right)\right|$ dataset reported by Fischer [44] and Xu et al. [60]. In Figure 6, we show the results of the fit of this dataset to Equation (15).

It is interesting to note that the deduced δ=175 ±13 nm is in remarkable agreement with its counterpart deduced for samples fabricated by bronze technology. The deduced parameters also have low dependence (~0.87), which is an additional indication that our model (Equation (15)) is not over-parameterized.

### 3.3. Samples Fabricated by Flükiger et al. by Bronze Technology [41]

Flükiger et al. [41] reported full $\left|\vec{F_{p}}\left(B\right)\right|$ curves, which we analyzed in Figure 1, for four samples fabricated using bronze technology. It should be noted that this research group utilized a different normalization procedure for the absolute value of the pinning force density from that used by other research groups [44,46,47,48,49,50]. Therefore, we analyzed this dataset separately (Figure 7). Although this dataset has only four $\left|F_{p,max}\left(d\right)\right|$ data points, we fitted this dataset to Equation (15) to estimate the robustness of our approach for extracting the characteristic length, δ, from limited $\left|F_{p,max}\left(d\right)\right|$ datasets. The deduced δ=146±15 nm is in the same ballpark as the δ values deduced from the fits to Equation (15) for large datasets (Figure 5 and Figure 6).

## 4. Discussion

The primary result of our analysis is that Nb_3_Sn conductors exhibit a fundamental length constant, δ, which is in the range of 146 nm ≤δ≤ 175 nm, and which characterizes the maximal intrinsic in-field performance of real world multifilamentary Nb_3_Sn-based wires.

Our current understanding of this unexpected result can be explained by two hypotheses, both of which are based on the interpretation that one of the two multiplication terms in the formal definition of the pinning force density (Equation (1)), $\vec{F_{p}}\left(J_{c},B\right)=\vec{J_{c}}⊗\vec{B}$, exhibits exponential decay with characteristic length δ. Thus, there are two possible scenarios/mechanisms.

### 4.1. Exponential Dependence of the $\left|\vec{J_{c}}\right|$ vs. Grain Size at $\left|F_{p,max}\right|$

This interpretation is based on an analog to the exponential decay $\sim e^{-\frac{x}{\lambda }}$ (more accurately $\sim \frac{cosh\left(\frac{x}{\lambda }\right)}{cosh\left(\frac{d}{\lambda }\right)}$ dependence, where d is the slab half-thickness and the layer thickness λ is the London penetration depth [58]) of the self-field transport current density from the superconductor–vacuum interface, which is London’s law. Considering that under high-field conditions, the interfaces in polycrystalline Nb_3_Sn are grain boundaries, we naturally came to Equation (15), in which the thickness of the layer (where the dissipative-free transport current flows at the condition of the pinning force maximum) is the characteristic length δ.

A schematic representation of δ-layers in the polycrystalline Nb_3_Sn phase, where we drew the δ-layer, is shown in Figure 8.

In this interpretation, large-size grains, d≫δ, are less effective areas for carrying dissipative-free transport current, because the central areas of these large grains do not contribute to transferring the transport current (Figure 8), and the current density is reduced by the exponential law. At the same time, small grains, d≤δ, are very effective areas for carrying dissipative-free transport current flow (Figure 8), because the full grain cross-section area works with approximately the same efficiency.

### 4.2. Exponential Dependence of the $\left|\vec{B}\right|$ vs. Grain Size at $\left|F_{p,max}\right|$

An alternative interpretation is based on an assumption that the flux pinning potential has exponential dependence $\sim e^{-\frac{x}{\delta }}$. As a result, the dissipative-free current can flow only within a thin layer (the thickness of δ) from both sides of grain boundaries, because the flux pinning is strong there and vortices can be held by the potential vs the Lorentz force. In this interpretation, central areas of large-size grains, d≫δ, also do not contribute to transferring the dissipative-free in-field transport current, because vortices are not strong enough vs. the Lorentz force. While the small-size grains, d≤δ, are very effective at carrying dissipative-free transport current flow (Figure 8), because vortices are pinned by pinning potential across the full grain area cross-section.

It is interesting to note that the schematic for the effective areas that can carry dissipative-free transport current is the same for both scenarios (Figure 8).

Thus, our current interpretation of the result is that the highest performance of the in-field transport current capacity of Nb_3_Sn wires is determined by the thin layer with a characteristic thickness of δ≅175 nm, which surrounds the grain boundaries from both sides.

It should also be noted that the maximum pinning force, $\left|F_{p,max}\left(J_{c},B\right)\right|$, represents the global maximum of the vector product of the transport critical current density,$\;\vec{J_{c}}$, and the applied magnetic field, $\vec{B}$, at any given temperature. In this study, we analyzed the $\left|F_{p,max}\right|$ values deduced from the $\left|F_{p}\left(B\right)\right|$ projection [38,39,40] of the $\left|F_{p}\left(J_{c},B\right)\right|$ curve. However, the same maximal values can be derived from the $\left|F_{p}\left(J_{c}\right)\right|$ [61] projections of the $\left|F_{p}\left(J_{c},B\right)\right|$ curve.

## 5. Conclusions

In this report, we reanalyzed experimental data on the dependence of the maximum pinning force density, $F_{p,max}$, from the average grain size, *d*, in practical low-*T*_c_ multifilamentary Nb_3_Sn conductors [1,2,3,4,5,6,7,8,9,10,11,12,13,14,15,16,17,18,19,20,21,22,23,24,25,26,27,28,29,30,31,32,33,34,38,39,40,41,42,43,44,45,46,55,56,58] fabricated by bronze and power-in-tube technologies.

The primary result of our analysis is that Nb_3_Sn conductors at their maximum in-field performance exhibit the characteristic length δ=175 nm, which is the same for samples fabricated by bronze and powder-in-tube technologies, which we interpreted as the characteristic thickness of the layer surrounding the grain boundary network where a dissipative-free transport current flows.

## Figures and Tables

**Figure 1 materials-16-05185-f001:**
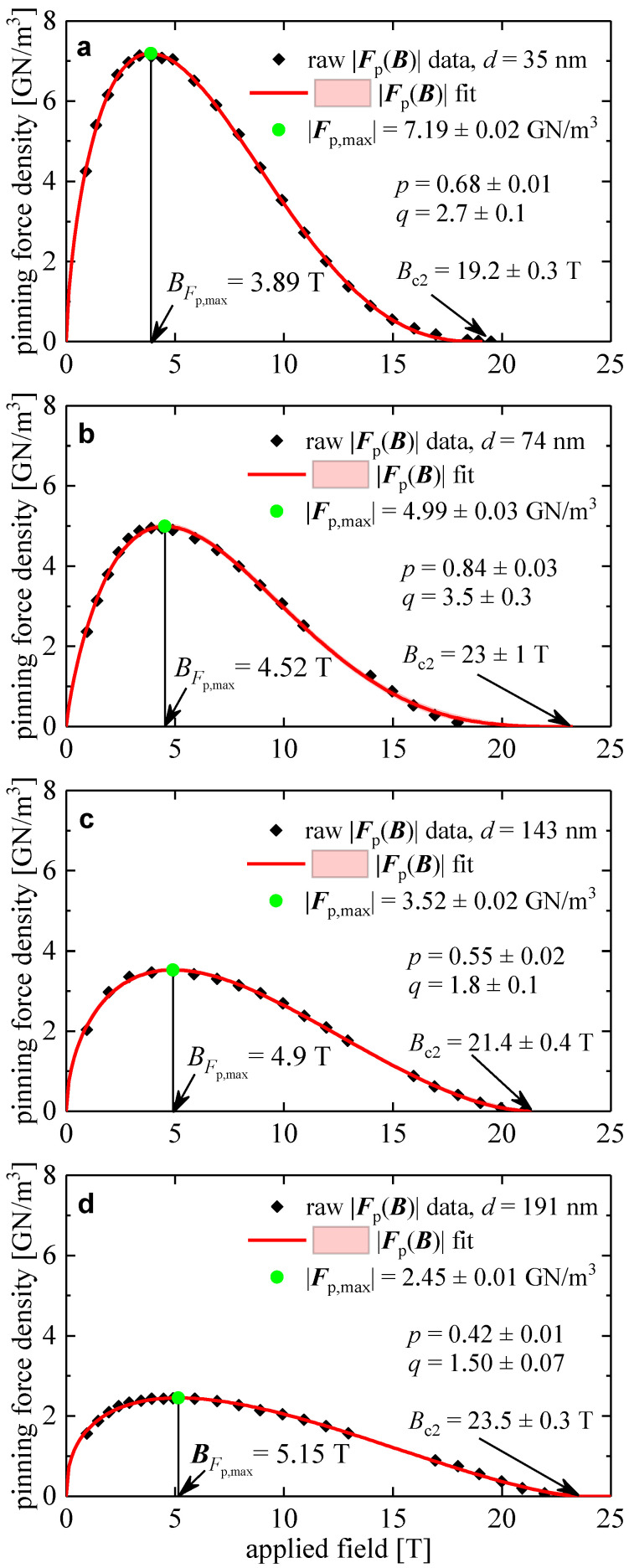
Pinning force density *F*_p_ versus *B* for bronze-route processed wires of different average grain sizes, *d*: (**a**) *d* = 35 nm; deduced *F*_p,max_ = 7.19 ± 0.02 GN/m^3^, *B*_c2_ = 19.2 ± 0.3 T, *p* = 0.68 ± 0.01, *q* = 2.7 ± 0.1; fit quality is 0.9997; (**b**) *d* = 74 nm; deduced *F*_p,max_ = 4.99 ± 0.03 GN/m^3^, *B*_c2_ = 23 ± 1 T, *p* = 0.84 ± 0.03, *q* = 3.5 ± 0.3; fit quality is 0.9982; (**c**) *d* = 143 nm; deduced *F*_p,max_ = 3.52 ± 0.02 GN/m^3^, *B*_c2_ = 21.4 ± 0.4 T, *p* = 0.55 ± 0.02, *q* = 1.8 ± 0.1; fit quality is 0.9987; (**d**) *d* = 191 nm; deduced *F*_p,max_ = 2.45 ± 0.01 GN/m^3^, *B*_c2_ = 23.5 ± 0.3 T, *p* = 0.42 ± 0.01, *q* = 1.50 ± 0.07; fit quality is 0.9986. The *p* and *q* parameters for the fit were determined using the Kramer–Dew-Hughes equation (Equation (2)). Raw data reported by Flükiger et al. [41]. The pink shaded areas show the 95% confidence bands.

**Figure 2 materials-16-05185-f002:**
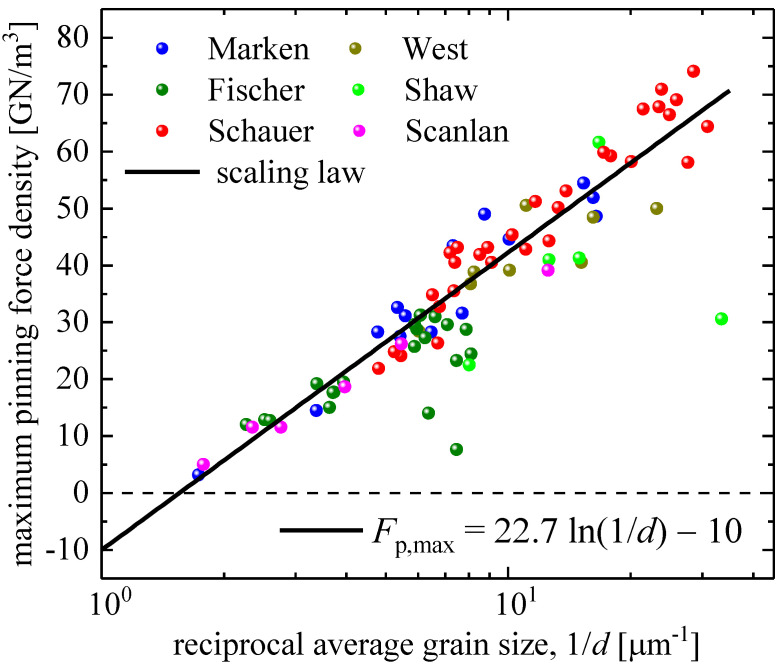
Maximum pinning force density, $F_{p,max}$, vs. reciprocal average grain size, 1/d, for datasets reported by Marken [46], West et al. [47], Fischer [44], Shaw [48], Schauer et al. [49], and Scanlan et al. [50]. Fitting curve (Equation (3)) was proposed by Godeke [45], who also presented the full dataset in a log–linear plot.

**Figure 3 materials-16-05185-f003:**
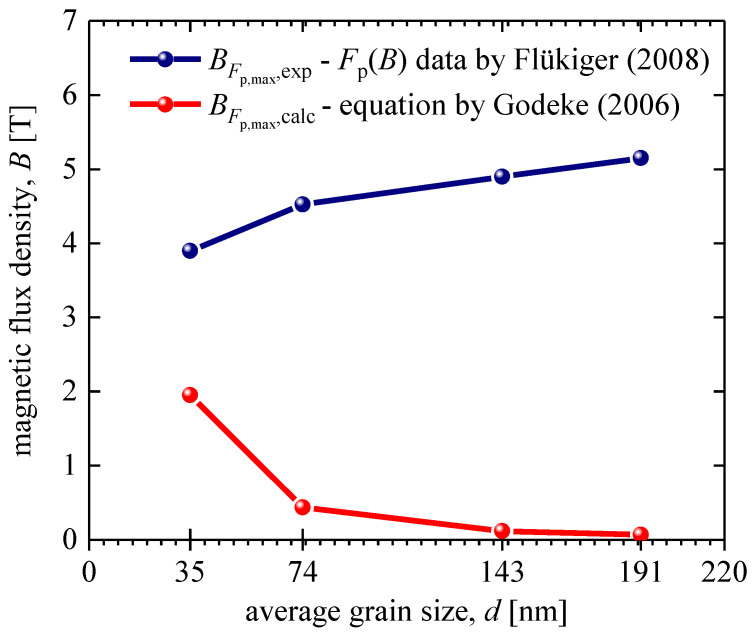
$B_{F_{p.max},calc}$ was calculated using Equation (4) (red) [45] and $B_{F_{p.max},exp}$ was extracted from experimental data reported by Flükiger [41] for Nb_3_Sn conductors fabricated by bronze technology.

**Figure 4 materials-16-05185-f004:**
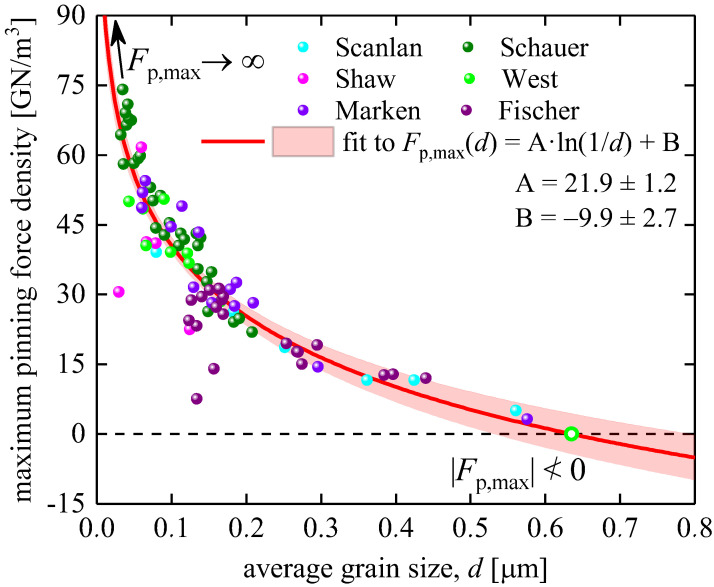
$\left|F_{p,max}\left(d\right)\right|$ data from Figure 2 (reported by Fischer [44] and Godeke [45]) in a linear–linear plot, and the fitting curve to Equation (3) [45], where we also showed both side extrapolations within the average grain size range of 20 nm≤d≤800 nm of Nb_3_Sn. Raw data reported by Marken [46], West et al. [47], Fischer [44], Shaw [48], Schauer et al. [49], and Scanlan et al. [50]. Pink shaded areas show the 95% confidence bands.

**Figure 5 materials-16-05185-f005:**
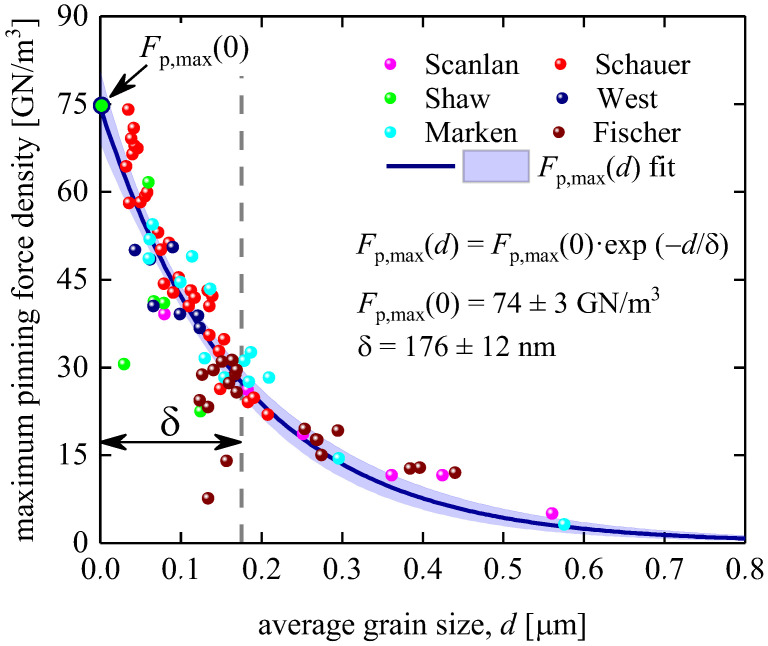
Maximum pinning force density, $\left|F_{p,max}\left(d\right)\right|$, vs. average grain size, d, for the non-Cu Nb_3_Sn wires and data fit to Equation (15). Raw data reported by Marken [46], West et al. [47], Fischer [44], Shaw [48], Schauer et al. [49], and Scanlan et al. [50]. Nb_3_Sn conductors were fabricated by bronze technology. Deduced parameters are $\left|F_{p,max}\left(0\right)\right|=74\pm 3\;\frac{\mathrm{GN}}{m^{3}}$, δ=176±12 nm; fit quality is 0.9248. Blue shaded areas show the 95% confidence bands.

**Figure 6 materials-16-05185-f006:**
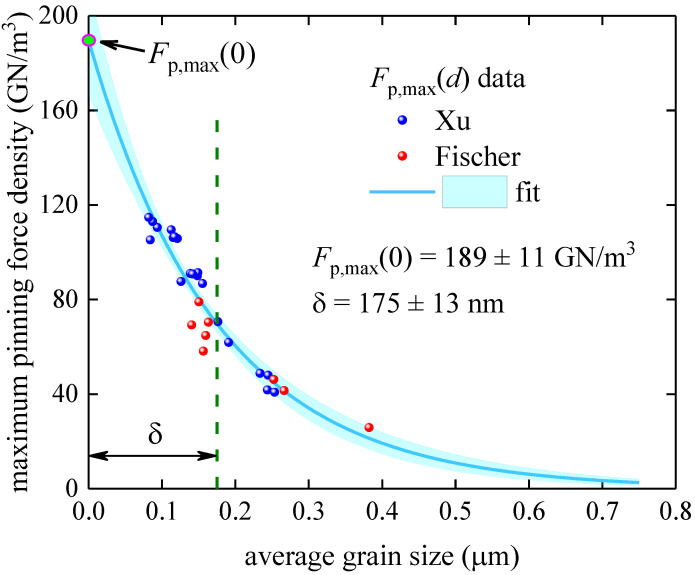
Maximum pinning force density, $\left|F_{p,max}\left(d\right)\right|$ vs. average grain size, d, and data fit to Equation (15) for the A15 layer fabricated by powder-in-tube technology [44,60]. Raw data reported by Fischer [44] and Xu et al. [60]. Deduced parameters are $\left|F_{p,max}\left(0\right)\right|=189\pm 11\;\frac{\mathrm{GN}}{m^{3}}$, δ=175±13 nm; fit quality is 0.9093. The cyan shaded areas show the 95% confidence bands.

**Figure 7 materials-16-05185-f007:**
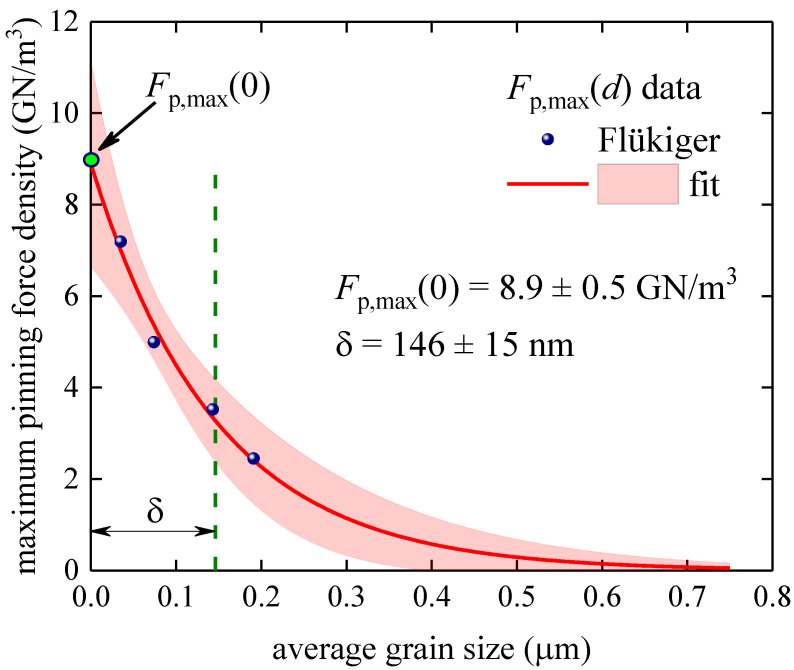
Maximum pinning force density, $\left|F_{p,max}\left(d\right)\right|$ vs. average grain size, d, and data fit to Equation (15) for samples fabricated by bronze technology and data fit to Equation (15). Raw data reported by Flükiger et al. [41]. Deduced parameters are $\left|F_{p,max}\left(0\right)\right|=8.9\pm 0.5\;\frac{\mathrm{GN}}{m^{3}}$, δ=146±15 nm. Fit quality is 0.9837. The pink shaded areas show the 95% confidence bands.

**Figure 8 materials-16-05185-f008:**
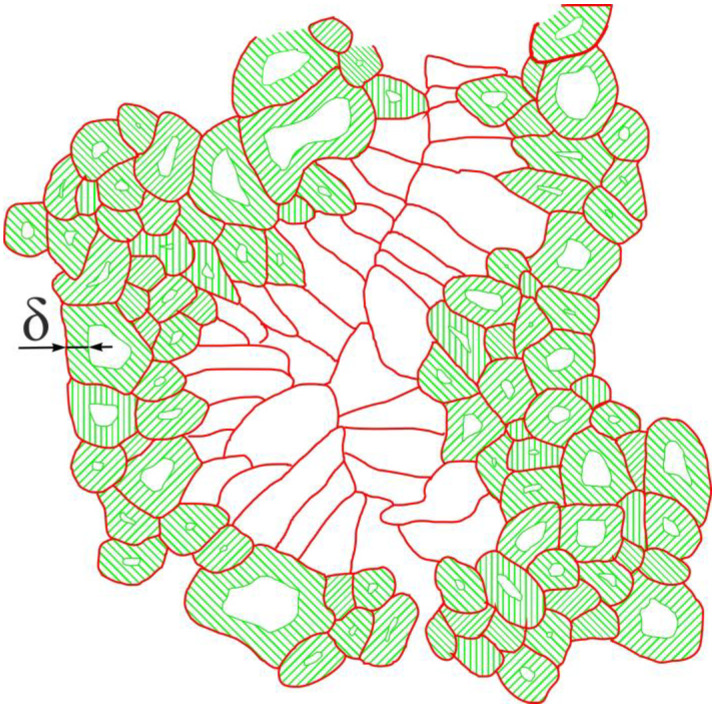
Schematic representation of the effective areas (δ-layer) in a cross-section of the equiaxed Nb_3_Sn layer.

## Data Availability

No new data were created or analyzed in this study. Data sharing is not applicable to this article.

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
