# Peer review of "Characteristic Length for Pinning Force Density in Nb3Sn"

_materials, 2023, doi:10.3390/ma16145185_

Round 1
Reviewer 1 Report
This study reanalyses experimental data concerning the dependence of maximum pinning force density on average grain size in multifilament Nb3Sn conductors at low temperatures T< Tc.
According to their analysis, Nb3Sn conductors exhibit a characteristic length delta =175nm at their maximum performance in the field. These results are the same for bronze and powder-in-tube samples. They interpret this thickness as the characteristic thickness of the layer surrounding grain boundary networks, where a dissipation-free transport current flows.
While this is a good idea, and efforts have been made to collect and analyze experimental results, some issues remain to be clarified:
-Abstract : The summary should be reformulated to prevent overloading it with equations, formulas and references. All these details could be covered in the text.
- Since equations 5 and 6 deal with the same variable, it would be preferable to replace them with a single equation.
-Equation 11 recalls the blackbody radiation law and the displacement of the maximum according to Wien's law. Why is this mentioned, and how does it relate to the present situation, which is quite different? If there is an analogy, it should be explained clearly.
-How do the authors explain the striking contrast between Godeke's (2006) and Flükiger's (2008) findings shown in figure 3?
- To validate the exponential law given in Equation 15, it would be simpler to take the logarithm, yielding a straight line. While the authors have directly employed the exponential form, maybe it's to mask discrepancies with experimental results (figure 5).
- The authors' results are available in this reference « doi: 10.20944/preprints202306.2153.v1 »... Is it a publication or just a reprint?
- Some portions of the manuscript are very similar to earlier work: DOI 10.1088/1361-6668/aab8dd, ....
It would be better to avoid this copy-paste, and to cite all resources used in the manuscript.
In conclusion, the manuscript needs to be revised in the light of the above comments and suggestions.
No comment on the English Language.
Author Response
Dear Reviewer,
Thank you very much for your prompt, detailed and professional review of our manuscript.
In response on that, we prepared point-by-point Response Letter and implemented all changes in the revised version of thee manuscript.
Tahnk you again.
The Authors.

Reviewer 2 Report
Talantsev and cowaorkers present a sort of review on the critical current in Nb3Sn superconductors. What they really do is reanalyse published data on the field dependent pinning force, and propose an updated model to describe the grainsize dependence of the maximum pinning force density.
The manuscript does a good job of justifying technological the need for higher critical current superconductors, and explaining how this requires a better understanding of the superconducting pinning.
They propose to replace a heuristic logarithmic function to describe the grainsize dependence with an exponential one, and then justify it with various fittings of a large amount of published data, and propose an understanding of this behaviour and the obtained length-parameter.
The study is well conceived and nicely presented, but for the following:
However, the authors make a remarkably strong/nasty criticism of the logarithmic function used in earlier research, and in particular of its use of "dimensioned" variables, highlighting this in the Abstract, and going so far as to recite a large number of otherwise important, but here utterly irrelevant, equations for superconductors (Eqs. 7-11).
Please note that it is a mathematical triviality to rewrite 22.7*ln(1/d)-10 --> 22.7*ln(0.64/d), where now 0.64um could serve perfectly well as a length scale, and is indeed identified in actual experimental data plotted in Fig.4. This part/aspect of the manuscript is offensive, galling, and preposterous.
overall the manuscript is well written, there are a few typoes, like Finning/Pinning, hold/held, etc. especuially towards the end (discussion/Conclusion)
Author Response
Dear Referee,
Thank you very much for your prompt, detailed and professional review on our manuscript.
In response on that, we revised the manuscript in accordance with your Comments, including the style and language in the Abstract and other parts mentioned by you.
Please find attached point-by-point Response Letter where all answers on your comments are given together with highlighted part of the text where these changes were implemented.
Thank you again.
The Authors.
